# Examining the Triple Burden of Malnutrition: Insights from a Community-Based Comprehensive Nutrition Survey among Indigenous Tribal Children (0–19 Years) in the Western Ghats Hills of India

**DOI:** 10.3390/nu15183995

**Published:** 2023-09-15

**Authors:** Abdul Jaleel, N. Arlappa, K. Sree Ramakrishna, P. V. Sunu, G. Jayalakshmi, G. Neeraja, D. Narasimhulu, T. Santhosh Kumar, Senthil B. Kumar

**Affiliations:** 1Division of Public Health Nutrition, ICMR-National Institute of Nutrition (NIN), Hyderabad 500007, India; jaleel.cp@icmr.gov.in (A.J.); sreekrishna1966@gmail.com (K.S.R.); sunupv1@gmail.com (P.V.S.); drjayalakshmi96@gmail.com (G.J.); daraneeraja14@gmail.com (G.N.); narasimha.dphn@gmail.com (D.N.); nnmbsanthosh@gmail.com (T.S.K.); 2Department of Epidemiology Statistics, ICMR-National Institute for Research in Tuberculosis (NIRT), Chennai 600031, India; senthilbsenthil@gmail.com

**Keywords:** malnutrition, anaemia, diet, children, adolescent girls, tribal population, *Attappadi*, Kerala, India

## Abstract

This article presents findings from a community-based cross-sectional study conducted in *Attappadi*, Kerala, India, aimed at assessing the prevalence of the triple burden of malnutrition among indigenous children aged 0–19 years. Historically, the indigenous population in *Attappadi* has faced significant developmental challenges, including high rates of malnutrition, infant mortality, and neonatal mortality. This study revealed alarming rates of undernutrition among children aged 0–59 months, with 40.9% experiencing stunting, 27.4% wasting, and 48.3% being underweight. Adolescent girls also suffered from undernutrition, with 21% classified as underweight and 43.3% experiencing stunting. Surprisingly, overweight or obesity was identified as a nutritional problem, affecting 1.4% of children aged 0–59 months, 4.2% of children aged 5–9 years, and 10.5% of adolescent girls. Additionally, a distressing proportion of young children aged 12–59 months (91.2%) were anaemic, with 50% diagnosed specifically with iron deficiency anaemia (IDA). Nearly all adolescent girls (96.6%) were reportedly suffering from anaemia. Deficiencies in vitamin B12, vitamin D, folate, and vitamin-A were prevalent among 35%, 20%, 16%, and 12% of children aged 12–59 months, respectively. The study underscores the urgent need for comprehensive interventions to address this triple burden of malnutrition. Recommendations include promoting culturally appropriate local food-based solutions, establishing participatory and community-led systems for health and nutrition information dissemination, and strengthening the nutrition surveillance system through village-level health and nutrition workers. By adopting a holistic approach, these interventions can help improve the nutritional status and well-being of the indigenous tribal children in *Attappadi*.

## 1. Introduction

The health disparities experienced by indigenous populations worldwide are deeply rooted in multiple factors, including geographical isolation, discrimination, and the loss of autonomy over their lands and culture [1,2]. These factors have had a significant impact on their social determinants of health, resulting in higher rates of poverty, undernutrition, marginalization, limited access to education and social services, and ultimately poorer health outcomes, increased disability rates, diminished quality of life, and shorter life expectancy compared to non-indigenous populations [3,4,5,6,7,8]. Indigenous peoples often confront pervasive racism and discrimination, which create additional barriers to accessing quality healthcare services, even when such services are available [2,9].

India is home to over 100 million tribal populations, constituting approximately 8.6% of the total population [10]. Despite the Constitution of India providing special provisions for tribal communities, they still encounter substantial challenges, rendering them one of the most underserved and undernourished segments of the Indian population [11,12]. Recent data highlight that 35.5% of children aged 0–59 months suffer from stunted growth, 19.3% are affected by wasting, and 32.1% are underweight. The situation is even more alarming for tribal children in the same age group, with stunting affecting 40.2%, wasting at 23.1%, and 39.4% experiencing underweight conditions [13]. The high level of growth failure profoundly affects their survival, growth, learning, school performance, and future productivity as adults. India’s ability to realise the Sustainable Development Goals (SDGs) requires addressing the needs of these most underprivileged and underserved populations.

The present study was conducted in *Attappadi*, a tribal region located in the Western Ghat hills in the state of Kerala. Kerala, a small and densely populated state in India has consistently ranked at the top in social development indices. The state has achieved remarkable progress in areas, such as life expectancy, female education, age at marriage, and mortality rates, including infant and child mortality when compared to other Indian states [14,15,16]. However, alongside its notable progress, the marginalization of indigenous tribes within the state remains a concern. Despite implementing numerous coordinated initiatives to address issues such as poverty, malnutrition, and infant mortality, the 4.2 lakh tribal population living in the state has not experienced commensurate benefits from the state’s socioeconomic advancements [17]. The tribal children in Kerala face a higher prevalence of stunted growth (33.2%) compared to their non-tribal counterparts (23.4%) [13]. Moreover, between the years 2012 and 2021, the tribal population in *Attappadi* witnessed a distressing number of 136 neonatal/infant deaths [18]. This figure is particularly concerning because Kerala generally boasts the lowest rates of neonatal and infant mortality (6 per 1000 livebirths) across the country.

Three major tribal communities—the *Mudugars*,* Irulars*, and *Kurumbars*—live in the region. These tribal groups collectively had a 30,332 population encompassing 3774 children (aged 0–6) spread across 192 hamlets [10]. However, despite being known as one of Kerala’s tribal heartlands, most of the population in *Attappadi* now comprises migrants from other places of Kerala and the nearest state of Tamil Nadu. The primary sources of livelihood for the tribal people in this area are agriculture, farm labour, and employment under the Mahatma Gandhi National Rural Employment Guarantee Program (MGNREGP). The region has two Primary Health Centers (PHCs), one Community Health Centre (CHC), and one Tribal Specialty Hospital to cater to the healthcare needs of the local population. Additionally, three Nutritional Rehabilitation Centers (NRCs) are functional to provide care and management of children (<5 years) with Severe Acute Malnutrition (SAM). There are also a few private health facilities in the area. To address hunger and ensure food security in the 192 tribal settlements, 172 community kitchens have been established.

To address the high prevalence of malnutrition and neonatal and infant mortality among the tribal population in *Attappadi*, the Indian Council of Medical Research-National Institute of Nutrition (ICMR-NIN) Hyderabad undertook a comprehensive nutrition assessment survey in 2022. The primary objective of this survey was to assess the nutritional status of various demographic groups, including young children (0 to 59 months), children (5–9 years), and adolescent girls (10 to 19 years). Furthermore, the study also aimed to evaluate the dietary practices (nutrient intake) of the population, recognizing the crucial role of nutrition in overall health and well-being.

## 2. Materials and Methods

### 2.1. Design and Sample Size

This community-based cross-sectional study was carried out by adopting quantitative and prevalence approaches. The sample size for this study was determined using the prevalence of underweight (48.5%) among children aged 0–59 months, as observed in a previous nutritional assessment conducted by the ICMR-NIN in *Attappadi* in 2013. The sample size calculation was performed considering a 95% Confidence Interval (CI), 20% relative precision, a 15% non-response rate, and a design effect of 2. Based on these considerations, the estimated sample size was 340 children (0–59 months). However, in anticipation of the potential refusal to provide blood samples for micronutrient analysis among children (12–59 months), the sample size was increased to 400. Additionally, data were also collected from children (5–9 years), and adolescent girls (10–19 years) in households with at least one 0–59 months child.

### 2.2. Selection of Villages and Households

The research team obtained the list of all the villages under the Integrated Tribal Development Project (ITDP) of *Attappadi* from the Project Officer. A total of 20 villages were randomly selected for the purpose of the study. Among these villages, 10 reported cases of infant deaths, while the other 10 reported no infant deaths between 1 January 2019, and 1 April 2022. Before collecting data, the research team conducted house listing in each selected village to identify households with at least one child aged between 0 and 59 months. The house listing activity was undertaken in collaboration with the Anganwadi Worker (AWW) from the Integrated Child Development Services (ICDS), and the Accredited Social Health Activist (ASHA) assigned to the respective villages. Using a simple random sampling method, the research team selected 20 households from the house listing of each selected village, ensuring that each select household had at least one child aged between 0 and 59 months (See Figure 1).

### 2.3. Data Collection

Data collection was conducted from May to June 2022 by a team of seven members consisting of two social scientists, an anthropologist, a nutritionist, a nurse, a biochemist, and a lab technician. Prior to the survey, all team members underwent standardized training in survey methodology, ethics, and tools which covered collecting socio-economic and demographic information, anthropometric measurements, 24-h dietary recall survey methodology, and blood sample collection and processing. The training focused on achieving maximum intra and inter-individual agreement for all measurements. To ensure the quality of data collection, a senior team member monitored the process. During the survey, the height/length and weight of all the available children in the selected household were measured and recorded using standardized instruments and methods. The Seca 813 digital scale and Seca 213 stadiometer were used to measure weight and height, respectively. The Seca 417 infantometer was used to measure the recumbent length of children under two years or height/length less than 85 cm.

A diet survey was conducted among 25% of the selected households to gather data on the diet of household members using the 24-h dietary recall method. This method involved a structured interview to obtain detailed information on all foods and liquids consumed by household members in the past 24 h, from midnight to midnight the previous day.

The haemoglobin status of children (12–59 months), and adolescent girls was measured using the Haemocue method. For this, a finger prick blood sample of 20 µL was collected and dropped on a Hb strip. To estimate the sub-clinical micronutrient status among children aged 12–59 months, 5 mL of venous blood was collected into a vacutainer tube. After collection, the vacutainer tubes were marked with the child’s ID and placed in cool boxes without direct contact with the ice packs. The samples were transported to the CHC in *Attappadi* at appropriate temperatures for further processing. At the CHC, the blood samples were centrifuged for 12 min at 2500 rpm and aliquoted into 1.8 micro litter storage vials for laboratory testing. Serum samples were stored in the refrigerator at −20 °C. Blood samples for serum retinol were stored in amber-coloured storage vials (1.8 microlitres) and covered with aluminium foil to protect from light exposure. The samples were then transported to the ICMR-NIN laboratory, Hyderabad, India, for estimation of micronutrients. The ABBOTT ARCHITECT I-1000 SR IMMUNO ANALYSER (Abbott, IL, USA) was used to test for Vitamin B12, folate, serum ferritin, and Vitamin D. Serum Retinol (vitamin A) was estimated using High-Performance Liquid Chromatography (HPLC) method.

### 2.4. Ethics

The study was approved by the Scientific Advisory Committee (SAC) of the ICMR-NIN, Hyderabad, India, and ethical clearance was obtained from the NIN’s Institutional Ethics Committee (IEC). Additionally, regulatory permissions were obtained from the Department of Health, Government of Kerala, and the Project Officer, ITDP, *Attapadi*. Written informed consent was obtained from the parents of the selected children aged 0–59 months. Similarly, assent was taken from children aged 5–19 years and their parents.

### 2.5. Data Analysis

All statistical analyses were conducted using Stata-14, a software package developed by Stata Corp. (College Station, TX, USA). Descriptive statistics were utilized to determine the prevalence of malnutrition. To evaluate malnutrition in children (0–59 months), Z-scores for height-for-age (HAZ), weight-for-height (WHZ), and weight-for-age (WAZ) were generated using the WHO-Anthro software (WHO-Anthro 3.2, Geneva, Switzerland). Z-scores for BMI-for-age (BAZ), and height-for-age (HAZ) were calculated for children (5–9 years), and adolescent girls (10–19 years) using Anthro-plus software. The WHO child growth standard cut-off values were used to classify children and adolescent girls into different categories of nutritional status [19,20]. The WHO haemoglobin cut-off values [21] were used to determine whether children and adolescent girls were anaemic or not. To classify children with iron, Vitamin A, Vitamin D, Vitamin B12, and folate deficiencies, we utilized the cut-off values provided by the World Health Organization (WHO) and the Institute of Medicine [22,23,24].

Individuals’ daily average consumption of various foods was estimated based on their age, gender, physiological status, and physical activity level. The content of various nutrients in the foods consumed by the children was calculated using the Indian Food Composition Tables (IFCT) and Nutritive Value of Indian Foods (NVIF). The mean intake of foods and median intakes of various nutrients were compared against the Recommended Dietary Intakes (RDI) for Indians and Recommended Dietary Allowances (RDA) for Indians, respectively, as recommended by the ICMR Expert Committee [25,26].

## 3. Results

### 3.1. Socio-Economic Characteristics of the Study Households

The survey included 523 children aged 0–59 months, 48 children aged 5–9 years, and 150 adolescent girls aged 10–19 years, from a total of 480 households. As we specifically targeted households that had at least one child falling within the 0–59 months age range, the other age category children, namely children aged 5–9 years and adolescent girls aged 10–19 years were drawn from the same households. Hence, the number of participants in these age groups were relatively less. Table 1 presents the background characteristics of the selected households. The households were comprised of 77.5% Irula tribes, 13.3% Kurumba tribes, 9% Muduga tribes, and 0.2% other tribes, with 98.9% of the households belonging to the Hindu religion, and 1.1% being Christians. Among the households, the majority (74.2%) were nuclear families consisting of couples with children. On average, the study population had a household size of 4.5 individuals. About 33.1% of the households had no land, and 48.8% had marginal land of less than 2.5 acres.

Regarding housing, 49.2% of the selected households had Pucca dwellings, while Semi-pucca and Kutcha houses accounted for 30.4% and 20.4%, respectively. The primary sources of drinking water were streams (34.4%) and taps (44.6%). Most households (68.8%) used firewood as cooking fuel, and 87.1% had a separate room for the kitchen in their homes. Electricity was available in most households (83.7%), and 74% of the households had sanitation facilities in use.

The Mahatma Gandhi National Rural Employment Guarantee Programme (MGNREGP) was the primary source of income for about 89% of households. Almost all (95.8%) households were protected from food insecurity by the Public Distribution System (PDS). Of the households selected, 90.6% had at least one beneficiary of the Integrated Child Development Services (ICDS) program. The prevalence of alcohol and tobacco use (daily or at least once a week) among the population aged 15 years and above was 21.8% and 42.3%, respectively. The prevalence of alcohol use was higher in men (46.8%) than in women (1.9%), whereas tobacco use (mostly in the form of chewing) was almost similar in men (48.8%) and women (42.3%).

### 3.2. Anthropometric Malnutrition

Table 2 shows the prevalence of malnutrition among children aged 0–59 months in *Attappadi*. Among the children, almost half (48.3%) were under-weight (low weight-for-age), while more than a third (40.9%) suffered from stunted growth (low height-for-age), and over a quarter (27.4%) experienced wasting (low weight-for-height). Among the malnourished children, 16 out of every 100 were severely underweight, 13 had severely stunted growth, and eight had acute undernutrition (wasting). Boys had a higher rate of stunting and underweight than girls. Boys were also more likely than girls to experience wasting, with a prevalence rate of 29.3% versus 25.2%. Approximately 1.4% of the children are overweight or obese (WHZ > 2SD). The prevalence of thinness (low BMI-for-age) among children aged 5–9 years was 35.5%, while 4.2% of children in the same age group were classified as overweight or obese (BAZ > 1SD). Additionally, it was found that 14.6% of children aged 5–9 years were stunted (low height-for-age).

Figure 2 presents the prevalence of thinness and overweight/obesity among adolescent girls aged 10–19 years. The overall prevalence of thinness (low BMI-for-age) was 20.9%. Among young adolescent girls (aged 10–14 years), the rate of thinness was higher at 25.6%, compared to older adolescent girls (aged 15–19 years), where the rate was 12.5%. The prevalence of overweight/obesity among adolescent girls was 10.5%. These statistics indicate that out of every 10 adolescent girls in *Attappadi*, two had low BMI indicating thinness, and one was overweight/obese. The prevalence of stunting among adolescent girls was found to be 43.3%, with 37.8% observed among girls aged 10–14 years and 54.1% among girls aged 15–19 years.

### 3.3. Anemia and Micronutrient Deficiencies

Table 3 and Figure 3 show the prevalence of anaemia among the study participants. The prevalence of anaemia was 91.2% among children aged 12–59 months. Among the anaemic children, 5.2% had severe anaemia, 57.8% had moderate anaemia, and 28.2% had mild anaemia. The prevalence of anaemia was highest (97.7%) in the youngest children (12–23 months), and it decreased slightly as children got older. Moreover, severe anaemia was more common (16.7%) in the youngest children (12–23 months) compared to older children. Anemia was a severe public health problem among adolescent girls, with an overall prevalence of 96.6%. The majority of adolescent girls have mild (53.4%) and moderate (42.1%) anaemia. In summary, anaemia is a significant public health concern among children and adolescent girls in *Attappadi*.

The study examined the prevalence of micronutrient deficiencies among children aged 12–59 months. Iron deficiency was particularly prevalent among tribal children, with one in every two children affected. The highest rates were observed among children aged 12–23 months (64.6%) and 24–35 months (58.7%). Vitamin A deficiency (VAD) affected 11.8% of the children studied. Folate deficiency was identified in approximately 16% of the children, with the highest prevalence among those aged 47–59 months (19.5%) and 36–47 months (17.8%). Vitamin B12 deficiency was observed in over one-third of the children (34.6%). Regarding Vitamin D deficiency, one in every five tribal children had inadequate levels of Serum 25 (OH) (<12 ng/mL), while an additional 47.5% had insufficient levels of Serum 25 (OH) (between 12 and 29.9ng/mL). The highest prevalence of Vitamin D deficiency was found among children aged 47–59 months (25.8%) (See Table 4).

### 3.4. Food and Nutrient Intake of the Population

Figure 4 illustrates the percentage of RDA of different nutrients consumed by three distinct age groups of children. Nutrient intake is derived from the food groups consumed by these children over the past 24 h. These food groups include cereals and millets, pulses and legumes, green leafy vegetables, other vegetables, roots and tubers, nuts and seeds, fruits, animal-based foods (meat, fish, etc.), dairy products, fats, and oils, etc. The data reveal that the protein intake of children of all age groups exceeded the RDA, with percentages of 166.4%, 190%, and 167%, respectively. In contrast, the energy intake of children aged 1–3 years, 4–6 years, and 7–9 years were 64.1%, 72.4%, and 79.6% of the RDA, respectively, indicating insufficient calorie intake. The calcium intake was deficient for all age groups, with children aged 4–6 years having the highest intake at 31.5% of the RDA. The iron intake was also below the RDA for all age groups, with children aged 1–3 years consuming the most at 55% of the RDA. The intake of Vitamin A, Thiamine, Riboflavin, Niacin, Vitamin C, and Zinc were all below the RDA for children in the community. Folate intake was sufficient for children aged 4–6 years and 7–9 years but below the RDA for children aged 1–3 years and 4–6 years. In conclusion, the graph demonstrates that children in the community were not meeting the recommended nutrient intake for most essential vitamins and minerals, except for protein intake. It is noteworthy that the iron intake among children of all age groups was nearly 50% of the RDA, which corresponds to the prevalence of iron deficiency anaemia at 50%.

Figure 5 shows the percentage of RDA of different nutrients consumed by adolescent girls in the community. The data show that the protein intake was meeting the RDA for adolescent girls. Only adolescent girls of sedentary physical activity were meeting the RDA for energy, Folate, and Niacin. The intake of calcium, iron, vitamin A, thamin, riboflavin, vitamin C and zinc was deficient for adolescent girls irrespective of their physiological status.

## 4. Discussion

This nutrition survey is the first comprehensive collection of evidence concerning the prevalence of various forms of malnutrition among different age groups in the tribal population of *Attappadi*, Kerala. This survey gained valuable insights, shedding light on existing disparities and evaluating the dietary practices of this population. The findings from this study hold great significance, as they provide a foundation for informed policies and targeted interventions aimed at breaking the cycle of disadvantage and improving the health and nutritional well-being of indigenous peoples in Kerala.

This study reveals that a significant majority of the selected households (89%) benefit from legally guaranteed employment, ensuring a livelihood for at least two adult members for a minimum of 200 days. Additionally, an impressive 96% of households are protected from food insecurity, receiving free grains and cereals through the PDS. Moreover, the study shows that 91% of children under 6 years old are provided with supplementary nutrition through the ICDS. These findings highlight the effective measures in place to support livelihoods, combat food insecurity, and address the nutritional needs of young children. Additionally, the study reveals that there are monthly cash incentives available to all pregnant and lactating mothers, further supporting their well-being and that of their children. Moreover, in all tribal hamlets, a community kitchen has been established to provide two meals every day, ensuring access to food for the community members.

Despite all these efforts, there is widespread malnutrition among the tribal children in *Attappadi*. High rates of anthropometric malnutrition were observed among children aged 0–59 months, children aged 5–9 years, and adolescent girls aged 10–19 years. Anemia also showed a similar trend. Another crucial aspect to consider is the escalating trend of overweight or obesity among children. In summary, the stunting prevalence among children in Kerala is similar to that in Vietnam. However, when specifically considering indigenous tribal children in the study, their stunting prevalence is comparable to the average prevalence seen in Sub-Saharan African countries, where it stands at 41% [27,28].

Contrasting with the micronutrient data derived from CNNS for the overall population of children aged 12–59 months in India, indigenous children residing in *Attappadi* show elevated deficiency rates in iron (50% vs. 32%), vitamin B12 (35% vs. 14%), and vitamin D (20% vs. 14%). Nonetheless, the deficiencies in folate (16% vs. 23%) and vitamin A (12% vs. 18%) is relatively less in these children when compared to the general child population in India.

Children in the community were not meeting the recommended nutrient intake levels for several essential vitamins and minerals. The calcium, iron, Vitamin A, Thiamine, Riboflavin, Niacin, Vitamin C, and Zinc intake fell well below the RDA for children of all age groups. Furthermore, their energy intakes were insufficient to meet the RDA. Interestingly, the only macronutrient that exceeded the RDA was protein. Protein intake was meeting the RDA for adolescent girls. Only adolescent girls of sedentary physical activity were meeting the RDA for energy, Folate, and Niacin. The intake of calcium, iron, vitamin A, thamin, riboflavin, vitamin C and zinc was deficient for adolescent girls irrespective of their physiological status.

The study also found alarmingly high levels of anaemia and malnutrition among adolescent girls. These findings suggest a concerning lack of adequate preconception nutrition among women, highlighting the potential risk of growth faltering during the prenatal stage. As a result, interventions addressing nutrition before conception and during pregnancy are equally crucial, alongside postnatal interventions.

To comprehend the current situation of the disproportionately high burden of malnutrition among the tribal population in a state like Kerala, which is widely recognized for its social sector development and high rank in social indicators, it is crucial to investigate the sociological and cultural aspects of the indigenous population of *Attappadi*. This will help us understand the complex underlying factors contributing to the persistence of high rates of malnutrition. Our fieldwork experience has revealed that, apart from economic poverty, the tribal population in *Attappadi* struggles with profound social poverty, characterized by a lack of sufficient and dependable relationships. As a result, they experience exclusion and a diminished sense of belonging within the larger society. Indigenous tribal peoples often find themselves as minority populations within economically prosperous societies [2,29].

It is important to note that the World Health Organization’s investigation into health determinants now recognizes colonization as a prevalent and fundamental determinant of Indigenous health [30,31]. Colonization in the context of *Attappadi* refers to the process by which the non-indigenous people established control over lands inhabited by indigenous peoples. This process often involves the displacement, marginalization, and subjugation of indigenous communities and the imposition of non-indigenous systems of governance, health, culture, and economic activities. Colonization has disrupted traditional lifestyles, cultural practices, and social structures, leading to the erosion of indigenous knowledge systems, loss of land and resources, and marginalization within broader society. By recognizing the role of colonization as a determinant of Indigenous health, we can begin to address the underlying structural issues that perpetuate health disparities among tribal communities in Kerala. This includes addressing social poverty, promoting inclusive policies and programs, and fostering relationships that empower and support the indigenous population in reclaiming their cultural identity and accessing their rights to health and well-being.

Another significant issue that emerged during our interaction with the tribal leaders was the lack of cultural sensitivity to the local health system. According to them, the health system emphasizes strict compliance, and the community adheres to its requirements primarily out of fear of facing negative consequences for non-compliance. The tribal leaders expressed their perception that the health system fails to understand or appreciate the unique context of their daily lives. We have found consistent findings from other studies that underscore the prevailing issue of cultural insensitivity within the healthcare system. Consequently, the services these systems provide often fail to align with indigenous communities’ socio-cultural practices, beliefs, or aspirations [32]. Unfortunately, in their pursuit of enforcing compliance, these health systems disrupt indigenous people’s established life patterns and impose a new way of life [33].

Studies have indicated that the health and well-being of indigenous peoples can be effectively supported by incorporating four fundamental dimensions of equity-oriented health services. These dimensions encompass inequity-responsive care, culturally safe care, trauma- and violence-informed care, and contextually tailored care [9,34]. Furthermore, fostering strong partnerships with Indigenous leaders, agencies, and communities is of utmost importance to effectively implement and tailor these fundamental aspects to specific local contexts. It is essential that Indigenous health initiatives are grounded in the priorities and needs of the Indigenous population they aim to serve.

The current study has demonstrated a range of challenges that the indigenous population of *Attappadi* faces that stem from discriminatory attitudes, exclusion, and social and economic poverty. Climate change would further compound the existing vulnerabilities of the indigenous population, leading to substantial repercussions on their health and nutrition. This ‘compounding’ effect appears to be much more prominent for Indigenous communities than other populations [34].

To address the issue of the triple burden of malnutrition, the study recommended a multipronged strategy that includes identifying and promoting culturally appropriate and acceptable local food-based solutions. Conducting a qualitative exploration is crucial in examining the barriers and facilitators of appropriate feeding practices, advancing preconception nutrition, and collaboratively developing culturally sensitive and locally relevant intervention strategies. A participatory and community-led system is also required to mainstream health and nutrition information. It is also reiterated that the nutrition surveillance system which involves screening and detecting severe anaemia and severe acute malnutrition in children should be strengthened using the services of the village-level grassroots health and nutrition workers. These efforts collectively have the potential to tackle the complex challenge of the triple burden of malnutrition, consequently leading to a reduction in the high infant and child mortality rates in *Attappadi*.

## 5. Limitation of the Study

The selected households and populations in the current study show similar socioeconomic and cultural characteristics. As a result, applying statistical association techniques such as logistic regression has certain limitations. These limitations hinder a complete elucidation of the predictors of malnutrition among the study population. Therefore, additional qualitative research is necessary to comprehensively explore the underlying causes of malnutrition and its associated factors within the study population. Another limitation of this research relates to the inherent errors associated with the Haemocue method. While the Haemocue method offers practical advantages, such as ease of use in field settings where quick and precise haemoglobin measurements are required, its inherent limitations should be taken into consideration when interpreting the findings of this research.

## Figures and Tables

**Figure 1 nutrients-15-03995-f001:**
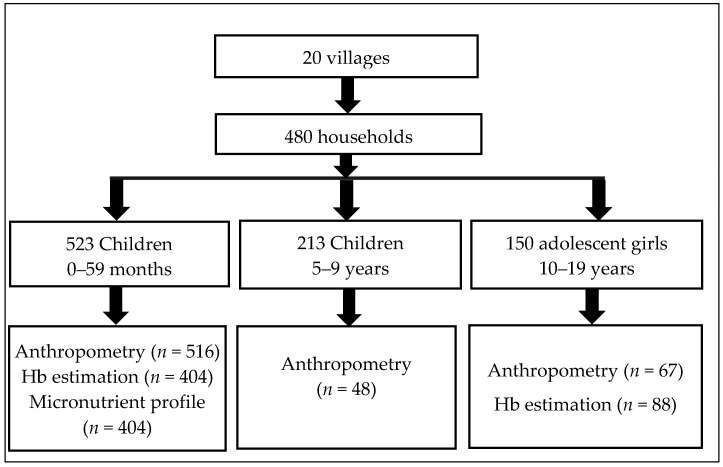
Details of the study villages, households, and participants.

**Figure 2 nutrients-15-03995-f002:**
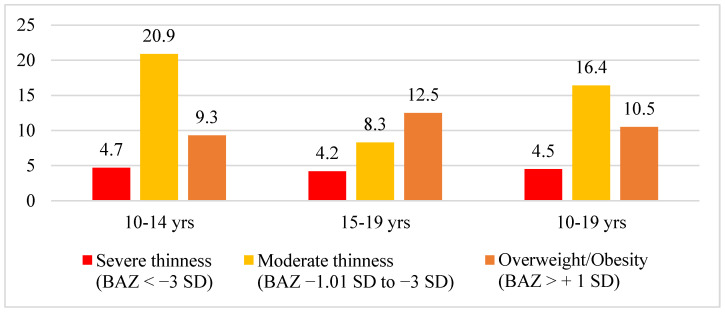
Prevalence of malnutrition among adolescent girls aged 10–19 years in *Attappadi*.

**Figure 3 nutrients-15-03995-f003:**
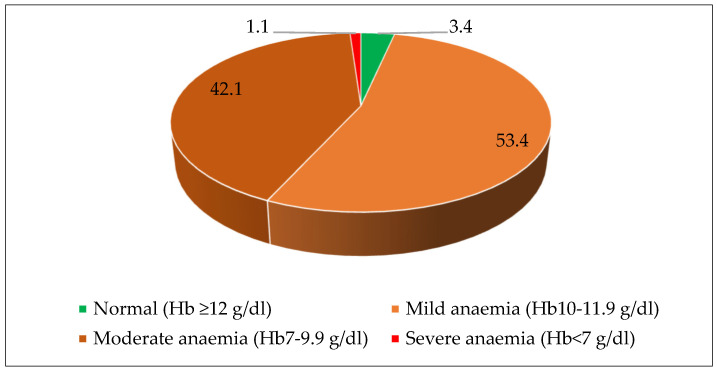
Prevalence of anaemia among adolescent girls aged 10–19 years in *Attappadi*.

**Figure 4 nutrients-15-03995-f004:**
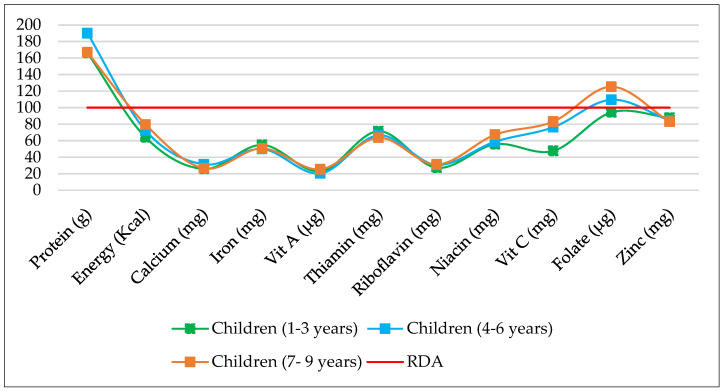
Intake of Nutrients (% of RDA) by children aged 1–9 years in *Attappadi*.

**Figure 5 nutrients-15-03995-f005:**
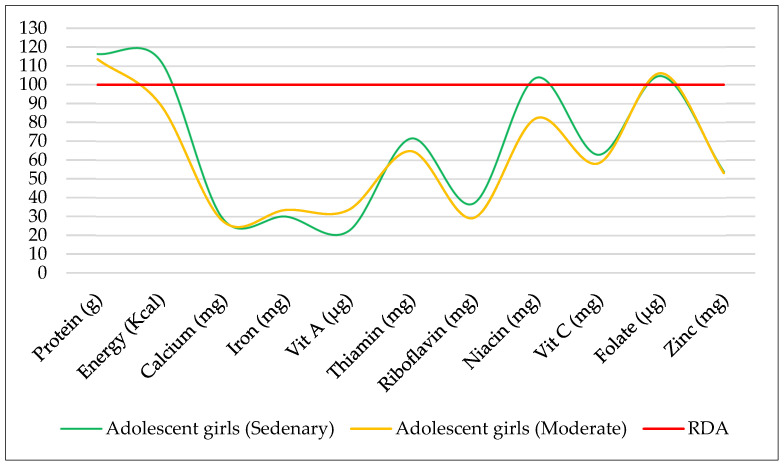
Intake of Nutrients (% of RDA) by adolescent girls aged 10–19 years in *Attappadi*.

**Table 1 nutrients-15-03995-t001:** Background characteristics of the selected households for the study.

Background Characteristics	*N* = 480	Per Cent
**Type of Tribe**		
*Irula*	372	77.5
*Kurumba*	64	13.3
*Muduga*	43	9.0
Others	1	0.2
**Religion**		
Hindu	475	98.9
Christian	5	1.1
**Type of family**		
Nuclear	356	74.2
Extended Nuclear	90	18.8
Joint	34	7.0
**Household size**		
2–3 members	101	21.0
4–6 members	345	71.9
7+ members	34	7.1
Mean household size [95% CI]	4.5 [4.4–4.6]	
**Landholding (Acres)**		
No Land	159	33.1
Marginal landholding (<2.5 acres)	234	48.8
Small landholding (2.5–5.0 acres)	51	10.6
Large landholding (≥5.0 acres)	36	7.5
**Type of house**		
*Kutcha*	98	20.4
*Semi-Pucca*	146	30.4
*Pucca*	236	49.2
**Source of drinking water**		
Open well	41	8.5
Tube well	20	4.2
Tap	214	44.6
Pond or tank	40	8.3
Stream/river/canal	165	34.4
**Cooking fuel used**		
Firewood	330	68.8
Liquefied Petroleum Gas	147	30.6
Kerosene	2	0.4
Biogas	1	0.2
**Availability of electricity**	403	83.7
**Availability of Sanitary Latrines**		
Present, and using	357	74.4
Present, but not using	48	10.0
No facility	75	15.6
**Availability of separate kitchen**		
Yes	418	87.1
No	62	12.9

**Table 2 nutrients-15-03995-t002:** Prevalence of malnutrition among children aged 0–59 months in *Attappadi*.

Weight-For-Age					
Age Group (Months)	*N*	Severe Underweight [WAZ < −3SD]	Moderate Underweight [WAZ −3SD to −2SD]	Total Underweight [WAZ < −2SD]	Normal [WAZ ≥ −2SD]
<12	77	11.7 [6.1–21.2]	26.0 [17.2–37.1]	37.7 [27.4–49.2]	62.3 [58.8–72.6]
12–35	209	16.3 [11.8–21.9]	32.5 [26.4–39.2]	48.8 [42.0–55.6]	51.2 [44.4–58.0]
36–59	230	16.2 [11.9–21.6]	35.4 [29.4–41.8]	51.5 [45.0–58.0]	48.4 [42.0–55.0]
0–59	516	15.5 [12.6–19.0]	32.8 [28.9–37.0]	48.3 [44.0–52.7]	51.7 [47.3–56.0]
**Height-For-Age**					
**Age Group (Months)**	** *N* **	**Severe Stunting** **[HAZ < −3SD]**	**Moderate Stunting** **[HAZ − 3SD to −2SD]**	**Total Stunting** **[HAZ < −2SD]**	**Normal** **[HAZ ≥ −2SD]**
<12	77	7.8 [3.5–16.5]	15.6 [8.9–25.7]	23.4 [15.1–34=3]	76.6 [65.6–84.9]
12–35	209	20.6 [15.6–26.6]	32.5 [26.5–39.2]	53.1 [46.3–59.8]	46.9 [40.2–53.7]
36–59	230	7.8 [5.0–12.1]	27.8 [22.4–34.0]	35.6 [29.7–42.1]	64.4 [57.9–70.3]
0–59	516	13.0 [10.4–16.2]	27.9 [24.2–32.0]	40.9 [36.8–45.3]	59.1 [54.7–63.2]
**Weight-For-Height**					
**Age Group (Months)**	** *N* **	**Severe Wasting** **[WHZ < −3SD]**	**Moderate Wasting** **[WHZ − 3SD to −2SD]**	**Total Wasting** **[WHZ < −2SD]**	**Normal** **[WHZ ≥ −2SD]**
<12	77	9.1 [4.3–18.8]	13.0 [7.0–22.7]	22.1 [14.0–32.9]	77.9 [67.0–85.9]
12–35	209	5.7 [3.3–10.0]	19.6 [14.7–25.6]	25.3 [20.0–31.7]	74.7 [68.2–80.1]
36–59	230	8.7 [5.7–13.2]	22.3 [17.3–28.2]	31.0 [25.3–37.3]	69.0 [62.7–74.7]
0–59	516	7.6 [5.6–10.2]	19.8 [16.6–23.5]	27.4 [23.7–31.4]	72.6 [68.6–76.3]

**Table 3 nutrients-15-03995-t003:** Prevalence of anaemia among children aged 12–59 months in *Attappadi*.

Age Group (Months)	*N*	Severe [Hb < 7 g/dL]	Moderate [Hb 7–9.9 g/dL]	Mild [Hb 10–10.9 g/dL]	Total [Hb < 11 g/dL]	No Anemia [Hb ≥ 11 g/dL]
12–23	84	16.7	66.7	14.3	97.7	2.3
24–35	115	2.6	67.0	26.1	95.7	4.3
36–47	95	2.1	50.5	33.7	86.3	13.7
48–59	125	2.4	48.8	35.2	86.4	13.6
12–59	419	5.2	57.8	28.2	91.2	8.8

**Table 4 nutrients-15-03995-t004:** Prevalence of micronutrient deficiencies among children aged 12–59 months in *Attappadi*.

Deficiency of Micronutrients		12–23 Months	24–35 Months	36–47 Months	48–59 Months	Pooled (12–59 Months)
Iron deficiency (Serum ferritin < 12 ng/mL)	*N*	65	109	90	123	387
	*n* (%)	42 (64.6)	64 (58.7)	38 (42.2)	49 (39.8)	193 (49.9)
Folate deficiency (Serum erythrocyte folate < 151 ng/mL)	*N*	73	109	90	123	395
	*n* (%)	11 (15.1)	11 (10.1)	16 (17.8)	24 (19.5)	62 (15.7)
Vitamin-B12 deficiency (Serum Vit B12 < 203 pg/nL)	*N*	70	106	90	122	388
	*n* (%)	24 (34.3)	34 (32.1)	31 (34.4)	45 (36.9)	134 (34.6)
Vitamin-D deficiency (Serum 25 (OH) concentration < 12 ng/mL)	*N*	74	109	91	124	398
	*n* (%)	13 (17.6)	17 (15.6)	17 (18.7)	32 (25.8)	79 (19.9)
Vitamin-A deficiency (Serum retinol concentration < 20 mg/dL)	*N*	74	109	91	124	398
	*n* (%)	10 (13.5)	11 (10.1)	12 13.2)	14 (11.3)	47 (11.8)

## Data Availability

Data described in the manuscript, and analytic code will be made available upon reasonable request.

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
