# Peer review of "Examining the Triple Burden of Malnutrition: Insights from a Community-Based Comprehensive Nutrition Survey among Indigenous Tribal Children (0–19 Years) in the Western Ghats Hills of India"

_nutrients, 2023, doi:10.3390/nu15183995_

Round 1

Author Response

We deeply value the insights provided by the reviewer, which have significantly contributed to enhancing the quality of our paper. We are pleased to inform you that we have diligently incorporated all the comments by the reviewer and meticulously revised the manuscript accordingly.

The point-wise details are attached for your reference. 

Reviewer 2 Report

1. The introduction would benefit from specifying the population size of children and adolescents in selected communities.

2. On table 4, the age groups presented have a discrepancy; they group children aged 36 to 47 and then they include in the next category 47 to 59.

3. In the discussion paragraph 3, they explain the results previously stated; it can be simplified to conclusions.

4. Throughout the article, they mention protein intake, vitamin intake, calcium, etc., but the sources of food are not clearly stated. It should be interesting to know this, as it is stated in the results adressing the sources of drinking water.

5. The discussion can be improved by comparing the results to more articles or research, preferably conducted in different contexts.

6. Lastly, I would like to congratulate the research team on this article. It takes into account all the resources the community receives and contemplates that even though there are a lot of policies, there is an important problem on malnutrition. It is a sensible approach to culturally diverse communities such as Indigenous people, and gives a global approach to this health disparity.

Author Response

(The authors gave the same response as above.)
